# Sustainable Supply Chain Management Practices and Sustainable Performance in Hospitals: A Systematic Review and Integrative Framework

**Verónica Duque-Uribe [1], William Sarache [1],\* and Elena Valentina Gutiérrez [2]**

[1] Departamento de Ingeniería Industrial, Universidad Nacional de Colombia, 170003 Manizales, Colombia; vduqueu@unal.edu

[2] Departamento de Ingeniería Industrial, Universidad de Antioquia, 050010 Medellín, Colombia; elena.gutierrez@udea.edu.co

\* Correspondence: wasarachec@unal.edu.co; Tel.: +57-68-879400

**Abstract:** Hospital supply chains are responsible for several economic inefficiencies, negative environmental impacts, and social concerns. However, a lack of research on sustainable supply chain management specific to this sector is identified. Existing studies do not analyze supply chain management practices in an integrated and detailed manner, and do not consider all sustainable performance dimensions. To address these gaps, this paper presents a systematic literature review and develops a framework for identifying the supply chain management practices that may contribute to sustainable performance in hospitals. The proposed framework is composed of 12 categories of management practices, which include strategic management and leadership, supplier management, purchasing, warehousing and inventory, transportation and distribution, information and technology, energy, water, food, hospital design, waste, and customer relationship management. On the other side, performance categories include economic, environmental, and social factors. Moreover, illustrative effects of practices on performance are discussed. The novelty of this document lies in its focus on hospital settings, as well as on its comprehensiveness regarding the operationalization of practices and performance dimensions. In addition, a future research agenda is provided, which emphasizes the need for improved research generalizability, empirical validation, integrative addressing, and deeper analysis of relationships between practices and performance.

**Keywords:** supply chain management; hospital supply chain; sustainable supply chain; hospital logistics; hospital sustainability; healthcare logistics; sustainable hospital management

## 1. Introduction

Supply Chain Management (SCM) is a field of growing academic interest, as reflected in the increase in related literature [1]. The Council of Supply Chain Management Professionals [2] (p. 187) not only defines SCM as the "planning and management of all activities involved in sourcing and procurement, conversion, and all logistics management activities," but also emphasizes its role in the integration between players involved in the entire supply chain. SCM interest lies in its contribution to a competitive advantage, in terms of differentiation and the reduction of operating costs, especially in the current context of intense competition, globalization, and active consumer participation [3]. It is argued that better SCM results in superior performance, through the adoption of exemplary practices [4]. A wide range of publications support the existence of significant relationships between SCM practices and organizational performance, especially from the economic perspective [5–8].

Beyond the previously mentioned economic focus, a recent trend in SCM study points to the consideration of its link to sustainability, which incorporates the environmental and social dimensions,

for two reasons. First, global poverty, health, working conditions, and climate change indicators [9], among others, have aroused worldwide interest in the promotion of sustainable development, defined as "the development that meets the needs of the present without compromising the ability of future generations to meet their own needs" [10] (p. 41). Second, given that organizations are often responsible for both environmental and social problems including pollution and unacceptable working conditions, they also have a duty to help mitigate such effects, as well as contribute to economic development.

The concepts of the Triple Bottom Line (TBL) and Sustainable Supply Chain Management (SSCM) have become significant. The former was coined by Elkington [11], and aims to consider the economic, environmental, and social dimensions to be equally important, since the economy is fundamental to support society, but doing business can become unfeasible in a depleted global ecosystem. The latter refers to the inclusion of environmental and social dimensions in the conventional notion of SCM, as proposed by Seuring and Müller [12] (p. 1700), who define SSCM as, "the management of material, information, and capital flows as well as cooperation among companies along the supply chain while taking goals from all three dimensions of sustainable development, i.e., economic, environmental, and social, into account, which are derived from customer and stakeholder requirements."

Hospital supply chains are often confronted by several economic, environmental, and social problems. From the economic point of view, increasing healthcare expenditures demand greater efficiency in the delivery of services [13,14]. The Organization for Economic Co-operation and Development has estimated that hospitals account for approximately 40% of total health expenditures [15]. Between 30% and 40% of a hospital´s budget is dedicated to supply chain costs [16], which can be reduced by up to 8% through the use of best practices [17]. In addition, said best practices allow clinical personnel to focus on their core mission of caring [16].

Regarding the environmental dimension, hospital processes and services are intensive in terms of material, energy, and water consumption, generate significant amounts of waste (especially toxic waste, as compared to other sectors), and account for a large carbon footprint [14,18,19]. In England, for instance, the Sustainable Development Unit of the National Health Service has calculated that healthcare's footprint represents 39% of public sector emissions, from which procurement contributes 57%, energy contributes 18%, travel contributes 13%, and others account for 11% [20]. Moreover, acute services are responsible for the largest portion, which is approximately 50% of the total.

Social problems related to hospital supply chains are also tangible. From an internal perspective, although hospitals are large-scale employers, non-standard forms of employment are frequent, pay levels have decreased in comparison to other economic sectors, women are compensated worse and recognized less often than men, daily working hours exceed legal limits, and safety considerations are often neglected [21,22]. Work characteristics such as shift work and long working hours not only increase the likelihood of occupational accidents, and developing burnout and additional psychological stress than in other jobs [14,23–25], but also impact the quality of patient care [26–28]. From an external standpoint, hospitals have a deep impact on the population because health services influence, in one way or another, peoples' quality of life. Nevertheless, reported global problems include unsatisfactory health service coverage for the needs of the population, in terms of access and delivery [21].

Therefore, the goal of accomplishing the triple challenge of being more efficient, more environmentally-friendly, and offering better conditions to both workers and communities served, leads to the subjects of SSCM practices and sustainable performance. No matter the way that practices are defined, whether as organizational routines, rules, or standard procedures [29], best practices are linked to the objective of that which is recognized as superior by a majority [16]. In other words, poor performance can be considered a consequence of a lack of best practices [30].

Numerous publications demonstrate that SSCM is a field of increasing interest. As Carter and Washispack assert in a review, "we have reached a point of saturation" [31] (p. 242), in terms of appraising the structure and main themes of SSCM literature. However, specific relationships between constructs remain unexplored. Some empirical studies stress that SSCM practices and sustainable performance constructs have not been clearly or consistently defined [32–34]. Besides the primacy in the

study of operational and economic topics, the environmental dimension has been more often addressed than the social dimension [35–39]. Moreover, the integration of the three sustainability dimensions has not been sufficiently robust [39–41], and industry-specific issues have not been elucidated to the extent to which they could be [34,42–44].

Despite dramatic growth in the SSCM literature [31], this is not the case when delimited to hospital settings. Academic database searches yield results on hospital SCM or hospital sustainability, but almost none appear to address hospital SSCM as such. Therefore, both SCM and sustainability may be relevant for hospitals, but they have likely been addressed in a fragmented manner in the literature. To the authors´ knowledge, there are no existing reviews which address the intersection between hospital SCM and sustainability. Reviews focused on the healthcare supply chain [45–49] have not explicitly considered environmental and social issues, whereas reviews on hospital sustainability [50] have highlighted the environmental dimension.

In response, the aim of this article is to present a systematic review and an integrative framework for SSCM practices that can contribute to sustainable performance in hospital settings. Three research questions are specifically addressed: (1) What are the main SSCM practices applied by hospitals? (2) What are the main sustainability performance metrics used by hospitals? (3) How can the relationships between SSCM practices and performance be framed in the hospital setting?

This paper is organized as follows. In Section 2, the methodology is presented and explained. Section 3 discusses the main findings, considering two main components. The SSCM practices applied by hospitals and sustainable performance metrics used by hospitals. In Section 4, an integrative framework, derived from the systematic literature review, is developed. Section 5 examines future avenues for research. Lastly, a relevant set of conclusions are presented in Section 6.

## 2. Materials and Methods

In order to address the proposed research questions, this study is based on a systematic literature review. Contrary to narrative reviews, systematic reviews are characterized by their explicitness and transparency regarding the methods used to find reasonable evidence on a given topic [51]. In management, it has been increasingly asserted that systematic reviews are a useful way to identify relevant scientific contributions, inform research and practice, and enhance a field's body of knowledge, by applying rigorous principles that have been traditionally used in medical research [52].

As shown in Figure 1, the methodology implemented to undertake this review involves three stages: planning, conducting, and reporting, which is in line with several suggestions [51–53]. The planning stage was accomplished through the identification of need, based on the research questions proposed, as well as through the definition of the search strategy, the selection criteria, the quality assessment criteria, the data extraction strategy, and the data synthesis approach. The conducting and reporting stages were accomplished from the contents of findings and discussion sections. In parallel, the Preferred Reporting Items for Systematic Reviews and Meta-Analyses (PRISMA) checklist [54] was also followed to ensure rigor of the review process.

Search strategy: Scopus and Web of Science (WoS) were selected for the search, due to their strengths in terms of extension, coverage, and the possibility of classifying sources in accordance with impact criteria [55]. For Scopus, publications throughout history, up to February 2019, were considered. For WoS, the time horizon was set between 2001 and February 2019, as the core collection of this database was available beginning in 2001. Based on the intersection between the topics addressed and the research questions, the executed string was as follows: (TITLE-ABS-KEY ("supply chain management" OR "healthcare logistics") AND practice AND hospital) OR (TITLE (sustainab* AND hospital)).

The term "performance" was excluded from the search string, as some publications only address practices, irrespective of their link to performance. Along with supply chain management, the term "healthcare logistics" was employed, considering that both have been used interchangeably [56]. Regarding the connection between sustainability and hospitals, as keywords cover broad and diverse

sustainability subtopics, the search was performed by the title, in order to ensure enhanced delimitation. The publications selected for this study were primarily in English, since the intention was to explore the topic globally [57]. Database search result duplicates were eliminated.

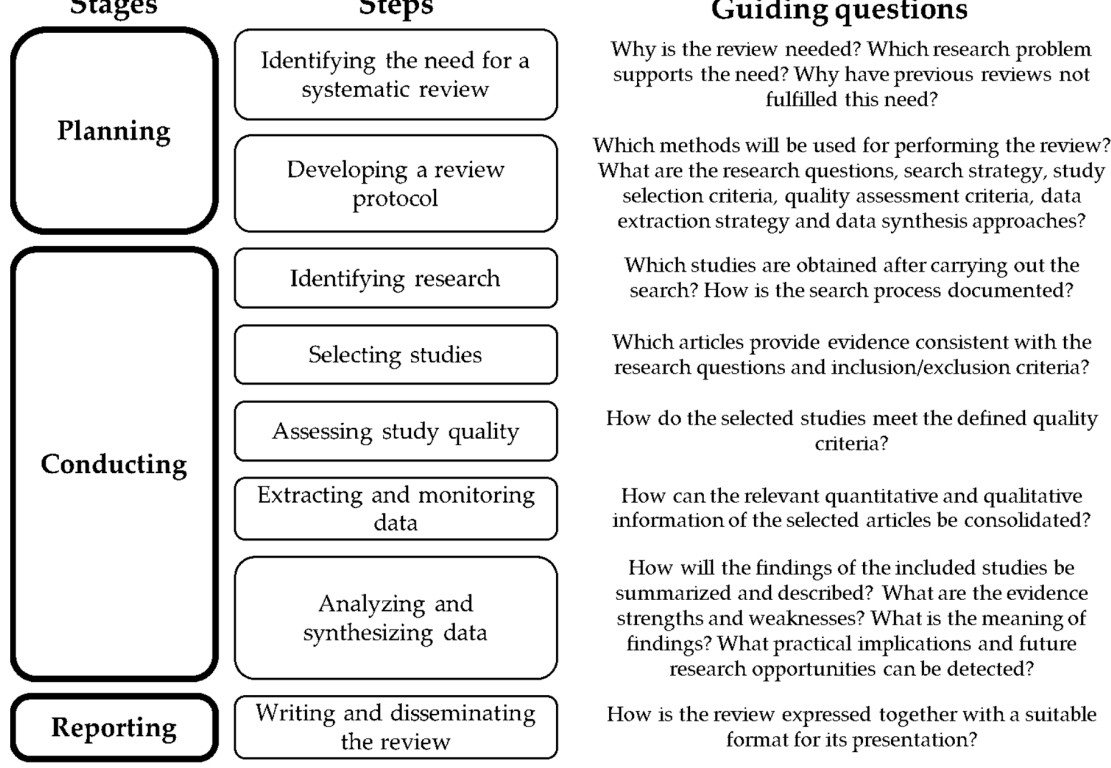

**Figure 1.** Methodology used for the review. Adapted from References [51–53].

Inclusion, exclusion, and quality assessment criteria: Publications with direct applicability to hospitals, from a comprehensive perspective, were included. Those that moved away from these entities as focal organizations, or on the contrary, focused on very specific chains, such as blood or laboratory, were excluded. Studies were also filtered based on their relationship to the TBL approach. Thus, contributions that referred to sustainability as the continuity of the specific health programs implemented, in order to analyze the effectiveness of such programs, were excluded. Articles and reviews from peer-reviewed journals were primarily considered. However, by review of publications' references, additional studies and international guidelines were considered suitable, such as References [58,59], since they specify SSCM practices applied in hospitals worldwide. Co-authors acted as coders to decide whether each publication retrieved from the search should be included or not. In cases of disagreement, these were discussed until consensus was achieved.

Data extraction strategy and synthesis approach: In accordance with the structure employed by most articles, as well as the information provided, the variables selected for data extraction, analysis, and synthesis were as follows: sustainability dimensions addressed, practices identified, performance metrics identified, and research suggestions. Concerning the data synthesis method, a mixture of interpretative and explanatory approaches was adopted, in an attempt to exceed description [51], as the pursued goal, being conceptual in nature, was the development of an integrative framework to facilitate the understanding of what, how, and why SSCM practices influence economic, environmental, and social performance in hospitals.

## 3. Results

As a result of the search strategy and selection criteria application, 58 documents were retained for analysis and synthesis (Figure 2). Out of 278 publications encountered in the databases, 80 were duplicates, 10 were added manually, and 150 were excluded, in accordance with the inclusion and exclusion criteria described in the methodology. This section is divided into three parts: in the first two, identified hospital SSCM practices and performance metrics are presented, respectively, in accordance with their categories. In the third section, specific practices and their effects on performance are discussed, to illustrate the relationships based on quantitative and qualitative findings reported in the reviewed literature.

| Search string | (TITLE-ABS-KEY ("supply chain management" OR "healthcare logistics") AND practice AND hospital) OR (TITLE (sustainab* AND hospital)) |
|---|---|
| Retrieved | +278 |
| Duplicated | - 80 |
| Excluded | - 150 |
| Sub-total | **48** |
| Manually added | + 10 |
| Retained | **58** |

**Figure 2.** Search strategy results.

### 3.1. SSCM Practices

Different approaches may be found in the literature regarding the concept of Hospital SCM. Reference [60] divides this concept into internal management (material and information flows), and external management (material, information, financial, knowledge flows, and relationships). This differentiation is also addressed by other authors, who refer to the concept of healthcare logistics. For example, Reference [61], based on Reference [62], identifies an external chain composed of manufacturers, distributors, purchasing groups, providers, and users, as well as an internal chain that includes supply management, inventory management, replenishment, and utilization. In addition to medical products, hospital logistics include the management of support services required for care. These may encompass food, laundry, patient transportation, information technology management, waste management, and home care services [63,64]. In fact, the concept of healthcare logistics is also meant to include operations such as care units and operating rooms [30].

In addressing the first research question, 13 categories emerged from this review for the classification of SSCM practices, which include: (1) strategic management and leadership, (2) supplier management, (3) purchasing management, (4) warehousing and inventory management, (5) transportation and distribution management, (6) information and technology management, (7) energy management, (8) water management, (9) food management, (10) hospital design, (11) waste management, and (12) staff and community behavior. A final category called "others" was designated to include contributions that were problematic to fit into the above-mentioned topics, despite their potential to provide important insights for practices. The rationale for establishing these categories emerged on examination of the ways in which practices have been classified previously in the literature, which Figure 3 illustrates. Some examples of practices, in accordance with the above-defined categories, are shown in Table 1.

**Table 1.** Summary of categories and examples of identified sustainable supply chain management practices in hospitals.

| Categories | Examples of Practices | Author(s) |
|---|---|---|
| Strategic management and leadership | 1. Establishment of a strategic plan for supply chain management. | [16,17,65,66] |
| | 2. Development of green and healthy policies and plans. | [58,59] |
| | 3. Executive support for supply chain management processes. | [17,58,59,61,67] |
| | 4. Use of indicators and measurement systems to assess total supply chain costs and performance. | [17,19,60–62] |
| | 5. Involvement of clinical and non-clinical staff in supply chain decision-making. | [17,65,68] |
| Supplier management | 1. Supplier base rationalization. | [62,65,69–71] |
| | 2. Sharing information with suppliers related to material flow management (forecasts, planned consumption, inventory, costs, promotions, and performance). | [46,60,72] |
| | 3. Inclusion of environmental, economic, and social dimensions in supplier arrangements. | [58,59,69] |
| | 4. Selection of ISO 14000-certified suppliers. | [58,59,69] |
| | 5. Work with suppliers to innovate and improve availability of sustainable products. | [58,59,69,72] |
| | 6. Assessment of suppliers´ sustainability and ethical practices. | [58,59,73] |
| | 7. Knowledge sharing and transfer (improvements, special handling requirements, good practices, technical issues, management solutions, and new product planning and development). | [60] |
| | 8. Payment control (enhanced control of payments to suppliers focused on preventing delays). | [60,65] |
| Purchasing management | 1. Supply standardization. | [17,46,62,65,70,74] |
| | 2. Use of purchasing groups. | [17,46,61,62,65,70,75] |
| | 3. Alliances between hospitals for the purchase of common items (aggregating purchasing volumes) to attain lower prices and avoid monopolies. | [46,64] |
| | 4. Use of the life cycle analysis to assess the environmental impacts of procured items. | [19,50,76] |
| | 5. Considering the environmental and human rights impact of procured products. | [58,59,73,77] |
| | 6. Purchasing of reusable, rather than disposable, products. | [50,58,59,76] |
| | 7. Eliminating, minimizing, and substituting chemicals for safer alternatives. | [58,59,78] |
| | 8. Coordination between hospitals to increase buying power for economic, environmental, and ethical purposes. | [58,59] |
| Warehousing and inventory management | 1. Determination of quantity to order and reorder points based on information systems. | [61] |
| | 2. Development of collaborative arrangements with trading partners to manage inventory of functional products (non-critical medical supplies) with high and stable demand (vendor-managed inventory, CPFR - collaborative planning, forecasting and replenishment, outsourcing). | [46,60,64,66,70,79,80] |
| | 3. Use of hybrid stockless systems (high-volume products are delivered directly to points of care and low-volume products are delivered to the central store). | [46,64,79,81] |
| | 4. Store consolidation and deployment of a centralized replenishment system for nursing units. | [16,62–65,74] |
| | 5. Deployment of a two-bin system. | [16,65,68] |
| Transportation and distribution management | 1. Consolidation of inter-site transport system. | [16] |
| | 2. Consolidation of external transport. | [16,70] |
| | 3. Promotion of active travel. | [50,58,59] |
| | 4. Promotion of public transport use. | [50,58,59] |
| | 5. Promotion of shared occupancy vehicle use. | [50,58,59] |
| | 6. Use of alternative fuels and technologies. | [58,59] |
| | 7. Development of services to minimize travel (e.g., telehealth, home healthcare, and videoconferencing). | [58,59] |
| Information and technology management | 1. Use of information systems and technologies in interactions between hospital departments. | [17,60,65,67,82,83] |
| | 2. Internal joint initiatives regarding product availability improvement and logistics cost reduction. | [60] |
| | 3. Deployment of an e-commerce system. | [16,60,62,63,70] |
| | 4. Use of track-and-trace systems (e.g., barcodes, Radio Frequency Identification). | [16,18,46,60,63,66,67,70,84,85] |
| | 5. Collaboration among supply chain partners using pharmacy information systems. | [84] |
| | 6. Automated central stores. | [16,66] |
| | 7. Use of automated guided vehicle systems for the transportation of pharmaceuticals, meals, linen, waste, patient files, tests results, lab tests, blood samples, and non-stock purchases. | [64,65,68] |
| | 8. Use of supplier relationship management system for the interaction between hospitals and their suppliers. | [60] |

**Table 1.** *Cont.*

| Categories | Examples of Practices | Author(s) |
|---|---|---|
| Energy management | 1. Implementing initiatives for saving (e.g., conducting periodic audits, installing variable-speed drive fans for operating theatres, automatic lighting timers, and sensors, updating lighting to LED). | [19,50,58,59,76,78] |
| | 2. Use of alternative technologies (e.g., cogeneration – combined heat and power). | [58,59,78] |
| | 3. Shifting to cleaner fuels. | [58,59,78] |
| | 4. Applying Lean Six Sigma approach to optimize a hospital linen distribution system. | [18] |
| | 5. Implementing social marketing interventions (turning off machines, lights out when unnecessary, closing doors when possible). | [86] |
| Water management | 1. Implementing initiatives for saving (auditing usage, controlling leaks, installing flow restrictors and dual-flush toilets, use of drought-resistant plants, reclaiming water from services such as dialysis and sterilization, harvesting rainwater). | [50,58,59,78,87] |
| | 2. Switching from film-based radiology to digital imaging. | [58,59] |
| Food management | 1. Serving locally grown and organic food. | [58,59,88,89] |
| | 2. Integrating the nutritional care pathway, nutritional standards, and regional menu framework. | [90] |
| | 3. Purchasing sustainable products (rBGH-free, cage-free eggs, meat produced without hormones or antibiotics, certified organic and fair-trade coffee). | [58,59,89] |
| | 4. Identifying and working with small, local vendors to achieve healthy food goals. | [58,59,90,91] |
| | 5. Limiting meat consumption. | [58,59,92] |
| | 6. Applying tariffs to reduce prices for more sustainable choices (e.g., vegetarian meals) and maintaining higher prices for less-sustainable options (e.g., high-fat dishes). | [91,93] |
| | 7. Recycling (fat, oil, grease, cardboard, paper, batteries, plastic, aluminum, newspaper, and tin cans). | [58,59,88,93] |
| | 8. Composting. | [58,59,88,89] |
| Hospital design | 1. Flow-through design (design for product, information, and people flow). | [62,65,68] |
| | 2. Integrated nursing workstations. | [62,65] |
| | 3. Building and adapting facilities considering sustainability criteria (using safer materials, local and regional materials, locating hospitals near public transportation routes, planting trees on site, incorporating design components such as day lighting, natural ventilation, and green roofs). | [19,58,59,78,94,95] |
| | 4. Application of sustainability healthcare-building assessment tools (e.g., BREEAM, LEED, and CASBEE). | [50,94–96] |
| Waste management | 1. Addressing over treatment and implementing methods like social prescribing. | [58,59,97] |
| | 2. Development of processes that use less material and improved technology. | [67,78,83] |
| | 3. Proper segregation. | [58,59,78,98,99] |
| | 4. Recycling. | [58,59,78,98,99] |
| | 5. Use of alternatives to incineration. | [58,59,78] |
| | 6. Setting of criteria and procedures regarding reverse logistics. | [71] |
| | 7. Take back programs of pharmaceuticals for patients and communities. | [58,59,71] |
| | 8. Applying Lean Six Sigma. | [18,30,100] |
| Staff and community behavior | 1. Hire/train well-qualified supply chain professionals. | [17,61] |
| | 2. Encouraging critical thinking within the community to understand, adopt, and promote sustainability initiatives. | [50,58,59] |
| | 3. Education of staff and community on sustainability. | [58,59,71,72,91,93] |
| | 4. Joint initiatives with the community for disease prevention and environmental health. | [58,59] |
| | 5. Collaboration with stakeholders to address environmental problems and develop plans to improve sustainability. | [58,59] |
| Other practices | Quality management practices (quality policy, employee training, product/service design, supplier quality management, process management/operating procedures, quality data and reporting, employee relations). | [83,101] |
| | Patient flow logistics (cross-functional or cross-organizational teams, information technology support, format standardization for information sharing, meetings focused on both medical and inter-organizational integration issues, and application of lean and agile approaches). | [14,102–104] |

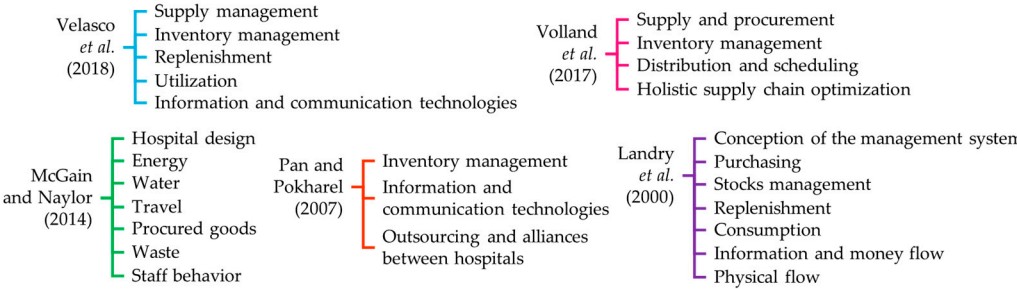

**Figure 3.** Categories identified for hospital sustainable supply chain management practices classification.

Strategic management and leadership practices are identified as a starting point to map out and control the resources, responsibilities, and implementation of other practices. Other categories, such as supplier management, purchasing management, warehousing and inventory management, transportation and distribution management, and information and technology management have traditionally been discussed from operational and economic perspectives. However, sustainability has contributed to the integration of environmental and social aspects into these topics, such as by considering the environmental and human rights impacts of procured products. Likewise, energy, water, food, hospital design, waste, and staff and community behavior have had a primarily environmental focus, as addressed by Reference [50].

"Other practices" comprises contributions that did not completely fit into the preceding categories. This holds true for studies that analyze the effect of quality management practices on non-financial and financial performance [101], and the application of an SCM perspective in patient flow logistics [14,102–104].

*3.2. SSCM Performance*

In addressing the second research question, SSCM performance metrics used by hospitals were identified and split into economic, environmental, and social factors (Table 2), as is customary in the SSCM literature [32,105,106], which is in line with the TBL approach. When applicable, metrics were also grouped into categories. For example, Reference [32] includes operation, market, and finance metrics as part of the economic dimension, pollution control, and resource utilization metrics as part of the environmental performance, and enterprise and employee perspective metrics as part of the social performance. Reference [105] divides performance into competitiveness, environmental, operations, and employee-centered and community social performance, while [106] groups metrics included economic, environmental, and social factors.

**Table 2.** Sustainable supply chain management performance metrics in hospitals identified in the literature.

| Dimensions | Categories | Metrics | Author(s) |
|---|---|---|---|
| ECONOMIC | Purchasing and supplier management | Categories of items handled, percentage of purchases using contracts, contract renewal, percentage of purchases using purchasing groups, number of employees dedicated to supply management, percentage of purchases made directly from manufacturers, percentage of purchases using consignment, level of sophistication of the purchasing planning process, total number of products per order, total number of purchasing orders, percentage of complete orders, percentage of urgent orders, number of indicators used in supply management, demand and forecast accuracy, delivery reliability, percentage of perfect orders delivered by suppliers, quick response, lead time from suppliers, and number of active suppliers. | [18,60,61,63,70,72,107] |
| | Warehousing and inventory management | Space utilization, order sorting, receiving completeness, cross-docking, service levels in the central warehouse (internal customers), inventory policies (manual/information system), number of Stock Keeping Units (SKU), order delivery (planned/not planned), number of indicators used in inventory management, inventory visibility, inventory availability, number of items in inventory, inventory levels, rupture rate, medical devices and pharmaceuticals stockouts, inventory accuracy, inventory turnover, reduction in stock variety, and reduction of time spent by clinical staff to perform logistics tasks. | [16,18,30,60,61,63,68,70] |
| | Transportation and distribution management | Perfect delivery condition, order delivery in full, delivery performance to customer commit date, on-time delivery, service speed, overall average delivery lead times for formal orders, urgent delivery, number of transactions (inputs-outputs), utilization of transport services, and medication delivery trips. | [18,61,63,67,70,72] |
| | Information and technology management | e-procurement (extent to which it is implemented), ease of use and usefulness, product identification, accurate and reliable tracking, information availability, information accuracy, information kept up to date, adherence to standards and rules, communication among parties, and amount of information sharing. | [18,63,72] |
| | Market | Market share, capacity to develop a unique competitive profile, market growth, market development, and market orientation. | [30,101] |
| | Processes and capacity | Perceived operation processes standardization, procedure preparation time and waste, service capacity, and increase in efficiency due to visual work standards. | [18,67,70,72] |
| | Financial | Purchasing costs for medical devices and pharmaceuticals, value of orders coming from tender processes, value of orders chosen without tender processes, administration costs for medical devices and pharmaceuticals flows, value of discounts and rebates, supply expense as a percentage of total hospital expense, supply expense per patient admission, supply expense per case-mix index adjusted admission, inventory value, value of inventory lost, inventory carrying costs and stocking requirements, transportation costs, revenue growth, profitability, net profits, return on investment, profit to revenue ratio, cash flow from operations, cash flow rate, share of net patient revenue, patient profitability, cost of services, operating costs, cost reduction due to the quality of service delivery, maintenance costs, savings due to efficiency and conservation improvements in energy, water, waste, travel, and food, and social investment volume. | [19,30,50,58–61,67,70,72, 86–88,91,92,98,101,107] |
| ENVIRONMENTAL | Procurement | Reduction of material consumption, drugs and packaging, decrease in consumption of hazardous/harmful/toxic materials, reduction in air emission/pollution from procurement, and reduction in air emission/pollution from anesthetic gases. | [50,58,59,78,100] |
| | Energy | Reduction in energy consumption, increase in energy efficiency, reduction in air emission/pollution from energy consumption, energy usage per unit area, and increase in the use of clean and renewable energy. | [18,19,50,58,59,78,86] |
| | Water | Water consumption, water footprint. | [50,58,59,78,87] |
| | Travel | Reduction in air emissions/pollution from business travel, patient transportation services, staff and community travel, increase in fully electric fleet and pool vehicles, reduction in fuel consumption, decrease in staff car use, and proportion of journeys made by a car. | [50,58,59] |

**Table 2.** *Cont.*

| Dimensions | Categories | Metrics | Author(s) |
|---|---|---|---|
| | Food | Percentage of locally and sustainably sourced foods procured, reduction in air emission/pollution from food supply, reduction in nutritional waste, and patient and staff satisfaction with healthy food choices provided. | [50,58,59,88–91,93] |
| | Hospital design and buildings | Compliance with environmental and social value certification standards. | [58,59,94] |
| | Waste | Decrease in waste generation from pharmaceuticals, chemicals, materials (e.g., products and equipment, packaging), and food, perceived waste reduction in processes, avoidance of improper waste mixing and incineration, proper waste disposal, percentage of toxic waste, decrease in incineration waste as a percentage of the total, improvement in ability to reuse/recycle/compost, and a reduction in waste disposal sent to a landfill. | [50,58,59,67,72,78,98,100] |
| SOCIAL | Quality of patient care | Death rate, timely provision of healthcare, length of stay, improvement in patient experience (quality of sleep, level of privacy, thermal comfort, service quality as perceived by customers, overall satisfaction with hospital experience), perceived care quality compared to other hospitals, service level, and perceived service level compared to other hospitals. | [30,60,72,83,86,101] |
| | Employee | Improvement in worker safety and health at work, improvement in employee awareness and education, improvement in worker efficiency, employee satisfaction, employee work life quality, proportion of working hours to that planned, staff absenteeism, employee privacy, and staff utilization. | [19,70,72,86] |
| | Community | Job creation, image/reputation among major customer segments, reduction in corruption and bribes, increase in population well-being, and stakeholder satisfaction. | [30,72,87,101] |

Findings show that most of the identified metrics are economic, which is coherent with the prominent attention that this dimension has received in the literature over time. Both recent and older publications that address the effects of SCM on performance, without explicit consideration of a holistic sustainability approach, have defined performance through a competitive advantage [7], operational [5] market, and financial constructs [108].

Conversely, the social dimension is that which contains the smallest number of identified indicators, which is consistent with the lesser recognition of this dimension in the literature [109]. On the one hand, social issues are considered difficult to measure, since they involve subjective, complex, and dynamic factors of human nature [110]. On the other hand, the literature shows significant advances in the identification of social issues of interest for supply chain management, but slow progress in their operationalization [109].

Environmental indicators are in a halfway position between economic and social indicators, pertaining to quantity. In the reviewed literature, efforts to measure natural resource consumption and waste generation, as well as the economic projection attributable to practice implementation, are evident. Predominance of this dimension over the social one may be explained by the considerable availability of publications and empirical results, mainly on green supply chain management [37,38].

*3.3. Analysis of SSCM Practices and Illustrative Effects on Sustainable Performance*

3.3.1. Strategic Management and Leadership

The SCM strategy has become a prerequisite for practice deployment [16]. However, it appears that strategy and organizational changes are hardly successful if there are no responsible and trained leaders who establish and control SCM priorities, plans, work teams, and performance measurement [17]. Clinical staff participation on logistics work teams is commendable, as it may help to solve natural conflicts between stakeholders [17,65], such as the product variety desired by physicians, in contrast to the economies of scale pursued by pharmacy managers [80]. To counteract patient demand uncertainty, high inventory levels are often seen as favorable by clinicians [17] and about 60% of supply spending

is influenced [74,111]. Thus, the incorporation of clinical perspectives facilitates a consensus about purchasing decisions, in order to reduce costs without detriment to quality.

From a sustainability perspective, which not only includes economic aspects, certain matters become relevant in advocacy for green and healthy hospitals. According to References [58,59], the organizational culture needs to be changed through practices like the development of green and healthy policies and plans, upper management and staff support for environmental and health issues, and the dedication of human and financial resources to green initiatives.

### 3.3.2. Supplier Management

It seems clear that organizational performance depends upon the way in which suppliers are managed [112,113]. Practices such as supplier rationalization are often suggested in the reviewed literature, as it lends not only the possibility of ordering higher volumes that generate financial savings, but also of building long-term relationships that enhance trust and enable the implementation of collaborative initiatives [65,69,70]. For example, Reference [70] suggests that the implementation of vendor-managed inventory arrangements is easier after having reduced the supplier base.

Reference [60] found that, in addition to having high levels of cross-departmental interaction, leading hospitals embark on joint initiatives with their suppliers to improve product availability and reduce costs, as well as extensively share information and knowledge with them regarding forecasts, consumption plans, inventory levels, costs, joint efforts, technical information, good material flow management practices, and new products and services. Concerning financial flow management, leading hospitals tend to keep payments under control, in order to prevent delays. As a result, these hospitals value the effects of this external integration positively.

Given that quality of care, health, and hospital reputation can be compromised by problems related to procured products. Hospital sustainability implies supplier sustainability as well. In manufacturing contexts, a typical example is when the procurement of harmful materials takes place, which can cause adverse events and lead to recall products from the market, as well as other consequences including criticism, damage to hospital reputation, and economic losses [114]. Therefore, the selection of certified suppliers [58,59,69], supplier sustainability reporting [73], supplier audit programs [62], and assessment of suppliers´ environmental and ethical practices [58,59,73] emerge as important practices in the arena of supplier management, as a way to ensure compliance with economic, environmental, and social standards. Furthermore, suppliers are uniquely poised to contribute to sustainability, as the development of more innovative and sustainable products largely depends on their capacity, readiness, and time invested therein [69].

### 3.3.3. Purchasing Management

Unsurprisingly, purchasing management is among the categories with the highest number of practices, as it represents a large portion of hospital budgets. On average, the share of supply expense, in reference to tangible supplies, is 15%, and can reach 40% in hospitals with high clinical complexity [107]. Among the most commonly mentioned practices are product standardization, purchasing group use, and creation of alliances with other hospitals. These practices have an essential economic orientation. For example, product standardization decreases item variety, and, therefore, the obtention of better prices and inventory reductions [62,65,70]. The use of purchasing groups has also turned out to be a beneficial practice for the achievement of more competitive prices and economies of scale, through the purchasing power acquired by these groups, as a consequence of volume consolidation [17,62,84]. Said effects also pertain to the creation of alliances with other hospitals [46,64].

Instead, practices like the consideration of the environmental and human rights impacts of procured products have more visible environmental and social backgrounds. Only in the United Kingdom does 57% of the healthcare footprint comes from procurement [20]. Consequently, criteria that refer to greater product durability, reduced waste generation, and less packaging and hazardous material use are recommended [58,59,75,77]. As Reference [75] demonstrates, purchasing of bundled

new and refurbished products may result in significant economic and environmental gains, if properly combined. Similarly, Reference [77] sheds light on how the packaging design needs to be considered in purchasing processes in order to improve logistic efficiency.

In addition to economic and environmental motives, the decision to source from suppliers that offer the most competitive prices must not be at the expense of unethical conditions and human rights violations, as in the case reported by Reference [58], in which ten-year-old children worked in the street to produce surgical scissors.

### 3.3.4. Warehousing and Inventory Management

Because about one-fifth part of healthcare revenue is attributed to inventory management [80], implementation of practices to improve the reception, warehousing, and control of supplies can be more than justifiable. The reviewed literature reports, for example, how the determination of quantities to order and reorder points based on information systems, in contrast to manual processes, has aided in the prevention of stock-out and overstock [61]. It further reports the ways in which the development of collaborative arrangements can be effective, depending upon contingent factors. As found by Reference [79], vendor-managed inventory is likely to work well for products with high and stable demand, which are not subject to highly-regulated environments, and when spatial complexity is low, the distance between organizations is not excessive and does not put supply at high risk of breakdown.

Similarly, the use of hybrid stockless systems has been recommended. This involves the delivery of high-volume products directly to points of care and low-volume products to central stores [64,79,81]. A completely stockless system is ideal for removing central stores and releasing space [68], but may fail in a hospital environment that deals daily with unpredictable emergencies, or in remotely-located hospitals in which response times might be significant [79]. Related to points of care, the findings of Reference [16] suggest that centralization of replenishment systems for nursing units is a practice that results in reducing surplus inventory, as well as administrative time for nursing, which works in favor of their dedication to delivery of care.

### 3.3.5. Transportation and Distribution Management

Case studies discussed by Reference [16] include practices related to transportation. In the Canadian hospitals explored by these authors, it was shown that the consolidation of inter-site transport produced economic savings of up to 35%, substantially reduced delivery times, and positively impacted customer service, which are a basis for the consolidation of external transport. Reference [70] emphasizes the need in the health sector for such external consolidation, as the large number of transport providers and their independent operations create valuable opportunities for capacity and routing optimization, which reduces both time and costs.

It is important to mention that a stream of practices pushes toward transport minimization for environmental reasons, given its high impact on $CO_2$ emissions [50]. From this perspective, the avoidance, or at least reduction of travel, is a primary goal, through the encouragement of active travel and the promotion of the use of public transport, shared occupancy vehicles, and electric vehicles [50,58,59]. Virtual solutions have proven valuable for the replacement of face-to-face meetings and appointments, since they avoid unnecessary patient and staff travel, both in administrative and clinical environments, through solutions such as tele-conferencing and telehealth, respectively [50,58,59]. In addition, the exploration of regulatory mechanisms, based on incentives and fees, is proposed, to stimulate the adoption of travel options with lower environmental impact and discourage those with the highest impact.

### 3.3.6. Information and Technology Management

Sharing information with suppliers was discussed in the subsection referring to supplier management. Some of the internal practices applied by leading hospitals mention the use of electronic communication tools and information systems, such as Electronic Patient Record (EPR), bar codes,

and Enterprise Resource Planning (ERP) systems [17,60,64]. The relevance of sharing information regarding forecasts, planning, inventory visibility, and delivery dates, as well as the establishment of cross-functional teams that encourage joint initiatives for product selection and standardization, inventory classification, and the discussion of performance metrics has been acknowledged [60].

A bundle of practices is concentrated on supply, inventory, and transport. These consider the implementation of electronic commerce or e-procurement, Radio Frequency Identification (RFID), the integration of medical and administrative information systems, and automation of warehouses and transportation systems. Some outcomes of e-procurement implementation include the reduction of clerical tasks, errors, use of paper, and associated costs [16,70]. RFID, along with barcodes, are part of track and trace systems, which identify medicines, individuals, supplies, or equipment. The identification of products, in particular, generates numerous advantages, in terms of a visibility increase and inventory cost reduction, manual task reduction, patient safety improvement, and support for reverse logistics [85]. By way of a case study about the location of infusion pumps with RFID, Reference [84] reached similar conclusions about the benefits of implementing this technology and even suggesting the integration of medical and administrative applications used by pharmacies to improve SCM agility.

Some experiences regarding the automation of central stores have come into being through the acquisition of carousels [16], while the use of automated guided vehicles has been suggested as a technology practice for transport [64,65,68]. Such vehicles are scheduled for the transportation of multiple items, such as pharmaceuticals, meals, linen, waste, patient files, tests results, lab tests, blood samples, and non-stock purchases. Although the investment payback has totaled approximately five years in hospitals that implemented automated guided vehicles, it has been considered a meaningful practice, given the minimal added value of conventional transportation jobs [68].

### 3.3.7. Energy Management

Several estimates provide notions of the high amount of energy consumed by hospitals. For example, it is calculated that these comprise 10% of total national consumption in the United States of America [18] and 20% of consumption in the Spanish tertiary sector [115]. Identified practices regarding energy mainly point to conservation measures, the use of alternative energy technologies and fuels, the application of lean six sigma, and behavior change interventions.

Motivated by facts such as the annual premature death of three million people due to air pollution, the University Health Network of Canada put a systemic approach into action that includes initiatives for energy use efficiency improvements. Some of these refer to the optimization of ventilation systems and replacement of existing lighting with LED, which resulted in quantifiable financial and consumption savings, and improved patient and staff comfort [19]. Similarly, Reference [76] suggests that variable-speed drive fans, lighting timers, and sensors have been effective in the reduction of energy consumption by up to 50% in operating rooms, while Reference [78] reports that cogeneration plants have allowed some hospitals to generate over half of their own energy.

By applying analytical tools derived from Six Sigma, Reference [18] proposed a future state to optimize a hospital linen distribution system, which led to improvements in communication, demand forecast accuracy, effectiveness, responsiveness, and reliability, which increased energy consumption efficiency. Reference [86] showed that turning off machines and lights when unnecessary, and closing doors when possible, as part of a social marketing intervention, proved successful not only in the reduction of energy consumption and carbon, but also in the improvement of the work environment and patient experience indicators such as quality of sleep and overall satisfaction.

### 3.3.8. Water Management

Hospitals use substantial amounts of water, which accounts for approximately 7% of the total water consumed in the tertiary sector in some countries [116,117]. According to the reviewed literature, auditing, controlling for leaks, and installing more efficient fixtures in both toilets and showers can



lead to savings of up to 25% [50,118], while more complex solutions might imply transformations in clinical services operation. The latter choice refers to options including switches from conventional radiology to digital imaging, which not only reduces water use, but also reduces harmful radiological chemicals [58].

Another focus of practices involves recycling water from sterilization, dialysis, and other processes [50,118] for use in non-potable needs [87]. To examine the impact of different policies related to water management in hospitals, Reference [87] proposes a causal model and studies two scenarios by using system dynamics. Simulation results indicate that a 15% water reduction policy leads to a reduction of 12% in the water footprint, savings in cost of services up to 14%, and a population well-being increases from 1.116% to 1.117%. In contrast, a 20% water reuse policy leads to a reduction of 16% in the water footprint, savings in the cost of services at 19%, and a population well-being increases from 1.116% to 1.117%. The water footprint denotes water consumption, cost of services refers to daily average cost of resources per patient, and population well-being is measured in terms of patient admittance.

### 3.3.9. Food Management

High-fat processed food, the use of non-nutritive additives, meat produced using antibiotics and hormones, obesity, antibiotic resistance, diabetes, cancer, food waste, and pollution caused by food transport are among the problems that current food systems face [88,89,91,92]. Hospitals have the potential to impact sustainability by addressing food issues, given their role as intermediaries in the market, their buying power, their responsibility for the promotion of proper nutritional habits, and the large number of people who frequent these organizations, between patients, visitors, employees, and the community [93].

In the reviewed literature, publications that focus on food sustainability show that recycling and avoiding the sale of bottled water are common practices, in contrast to composting and serving organic and locally grown food [88]. Reference [93] identifies 12 opportunities through which food practices may be addressed: procurement, catering contracts, menu development, pricing, waste management, infrastructure, staff training, information, education, communication and feedback, partnerships, and special events. Similarly, Reference [91] suggests practices that range from the participative design of new options with staff and customers to behavioral initiatives that encourage the consumption of healthier food, while Reference [92] shows that reducing meat consumption by up to 20% and substituting it for vegetarian or alternative proteins from local sources is feasible for hospitals, without a detriment to budgetary increases.

A case presented by Reference [90] provides insights regarding the improved fulfilment of patients´ nutritional needs, their increased satisfaction, waste reduction, and local economy enhancement, by sourcing from a single and local supplier and articulating nutritional standards with regional menu frameworks. However, unlike Reference [90], positive outcomes in all sustainability dimensions are sometimes mixed. For Reference [88], the implementation of food sustainability practices overrides their costs, whereas Reference [93] finds cost to be an obstacle. Reference [91] concludes that not only is price a restriction on healthier food, but so too is the difficulty of preparation, staff involvement, and creativity to promote said meal in a market that is accustomed to and satisfied with fries and sugar-sweetened beverages.

### 3.3.10. Hospital Design

For any hospital, fluid architecture is desirable to facilitate logistics, which, in turn, assists with people, material, and information flows [65,68]. Coherently, one of the practices implemented in some hospitals has been the integration of nursing stations, through a design that groups the elements of information, medicines, and materials required for care, and which not only contributes to ergonomic improvements, but also contributes to reducing the distances travelled by nursing staff [65].

In addition to making flows more effective, sustainability raises challenges that generate the need for more complex planning for future facilities, as well as adaptation of existing facilities [96]. These challenges are aimed to ensure more efficient use of resources such as energy, water, and waste management, better social conditions, in terms of accessibility, safety, comfort, and patient experience, and improved economic outcomes with a reference to life cycle costs and contribution to local economies [94]. Specific recommendations for building and adapting facilities, considering sustainability criteria, include using safer and local materials, sitting hospitals near public transportation routes, planting on-site trees, and incorporating design components like day lighting, natural ventilation, and green roofs [58,59].

One limitation of current sustainability demands, however, many of today's hospitals operate in old buildings that consume large amounts of resources, and whose design is not carefully planned to favor aspects such as those mentioned above [96]. Furthermore, trade-offs can arise between dimensions. For example, larger patient rooms generate greater comfort, but consume additional environmental resources [94,119]. An initial step for hospital building modification lies in the application of a sustainability assessment tool. For this purpose, different options, such as the Building Research Establishment Environmental Assessment Method (BREEAM), Leadership in Energy and Environmental Design (LEED) for Healthcare, Green Star Healthcare, and Comprehensive Assessment System for Built Environment Efficiency (CASBEE) have been developed [95].

### 3.3.11. Waste Management

Significant volumes of waste are generated by hospitals. In Victoria, Australia, for example, public hospitals generate as much waste as 200,000 households [76]. Besides environmental motives, waste management is important for public health reasons. In countries like India, Reference [98] found that regulation is still weak while non-hazardous and hazardous waste are often mixed together, and large amounts of waste are unnecessarily incinerated, which causes avoidable toxic air pollution. Consequently, Reference [98] proposes a system that encompasses reduction strategies, segregation, and recycling of non-hazardous waste. After conducting a pilot study at one hospital, they projected quantities that can be prevented from being improperly disposed and incinerated, in addition to the economic benefits that this would bring, together by increasing recycling.

Even more effective practices refer to avoiding waste generation altogether, which has been made achievable by addressing overtreatment, instigating methods such as social prescribing, development of processes where less material is necessary, waste stream analysis, review of waste-generation processes, selecting safer chemicals, purchasing environmentally-friendly products, purchasing reusable rather than disposable products, and acquisition of improved technologies [58,76,78,97]. Consequently, some hospitals have reported decreased use of hazardous chemicals like mercury, and reduced waste generation, which translate into financial savings [78]. As stated by Reference [76], single-use plastic trays double the cost of reusable trays, which means that only the reuse of these elements would represent annual savings of $5,000 for a 300-bed hospital.

Case studies regarding the application of Lean Six Sigma in medication processes [18,100] and sterile processing [18] illustrate the way in which it can lead to improved medication availability, fewer missing medications, reduced medication delivery trips, less kit variety for sterile processing, less waste, and financial savings. Similarly, despite the fact that waste cannot be completely avoided, Reference [71] demonstrates that reverse logistics processes offer significant opportunities for hospitals and healthcare systems as a whole. An intervention undertaken in one hospital allowed the value to return to stock, and it was found that recycled and disposed drugs represented around 3% of total drug expenditures. This was a starting point for establishing responsibilities, criteria, procedures, and schedules for the collection, review, and classification of returned items [71].

### 3.3.12. Staff and Community Behavior

Several examples show that capabilities, culture, and psychological factors are key determinants for the successful implementation of SCM practices. In other words, lack of training and education of supply chain professionals and executives is a common barrier [17,61], as is proneness to issues like sharing information, which pivots on organizational culture [17]. Variables like shared values have proven to play a mediating role between other variables, such as the use of electronic medical records and physicians´ performance [120], while willingness to support the implementation of practices depends on individual interests and the degree to which they make true sense for employees, patients, and the community [50].

From an economic perspective, Reference [66] includes the institutionalization of training and development as a best practice for the improvement of hospital supply chains. Reference [17] draws attention to the existing need for training in analytical skills, SCM best practices, leadership, communication, and financial themes, among others. From the environmental and social perspectives, Reference [50] asserts that critical thinking needs to be fostered, and employees must be supported in the process of making ethical decisions that they consider to be coherent with their personal beliefs, if indifference toward and myths about sustainability that constrain action are to be dismantled.

### 3.3.13. Other Practices

Several studies have found positive and significant links between organizational performance and quality management practices. Reference [101] highlights the importance of encouraging staff involvement, managers development, and strengthening information and statistics tools, since employee relations, training, role of top management, and quality data and reporting, were found to prevail over practices related to factors such as service design, supplier quality management, and process management. Another interesting result is the concluded influence of quality practices on financial performance through non-financial performance, which might indicate the pertinence of investment in quality practices even if it does not result in better financial performance in the short run, but indirectly through market share gains, increased service quality, and other outcomes. [83,101]. Such findings provide helpful evidence for prioritizing these kinds of practices and support of their implementation, since they contribute to market share gains, increased service quality, reduced waste, higher speed, improved quality of care, a superior competitive position, and financial performance.

Lastly, attention should be called to certain publications that are connected with the realm of operations management, which employ the SCM concept as a guide to study planning processes, which assumes that patient flow can be more efficient if a rationale similar to that of product flow is applied. References [103,104] suggest enhancing cross-functional or cross-organizational teams as well as information technology support, and format standardization for information sharing and meetings focused on both medical and inter-organizational integration issues, to address communication, patient safety, waiting times, and integration problems that arise when manifold healthcare providers are involved in patient care. Similarly, Reference [102] discusses the applicability of lean to SCM in combination with agility, while Reference [14] argues that a socio-ecological approach can be applied in hospitals by moving sustainability into the core business, which requires that decisions be made about care planning and service design.

## 4. A Proposed Framework for Hospital SSCM

In response to the third research question, this section presents an integrative framework for SSCM practices that may impact sustainable performance in hospital settings (see Figure 4). This can be considered innovative in at least three ways. First, as found in the reviewed literature, several publications have outlined relevant sustainability issues, but little attention has been given to the amalgamation of scattered practices and performance measures in a single and articulated framework. Most of the previous research on hospital supply chain management focuses on logistics from a cost

reduction perspective, which is indisputably crucial for sustainability, but is insufficient from the TBL approach. Moreover, the publications identified with the sustainability label pivot primarily on the environmental dimension and leave aside the social dimension.

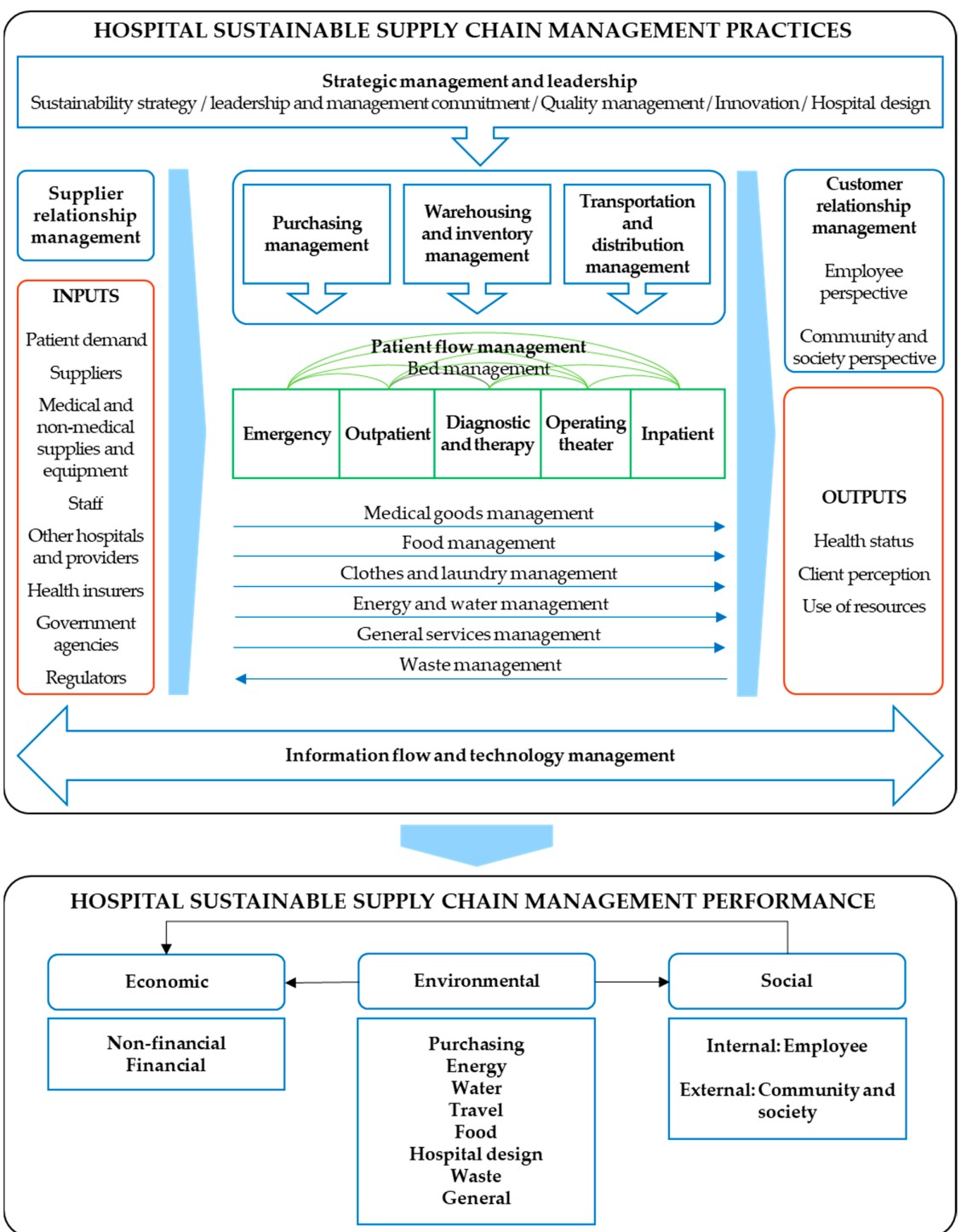

**Figure 4.** Conceptual framework for hospital sustainable supply chain management.

Second, the wide-ranging identification of practices and performance metrics achieved in the literature review, which gave rise to the proposed framework, likely allows to delineate a clear path toward empirical validation and the managerial implications of practice implementation and performance measurement. While a considerable number of frameworks provide valuable insights on

interactions in sustainable supply chain management [40], healthcare supply chain management in the emerging economy with the sustainable lenses [121], and supply chain sustainability in the service industry [122], the degree of operationalization of the categories and exemplified relations presented in this case has not been detected in previous reviews.

A third contribution to highlight is the worldwide applicability of the proposed framework and its possible extension to other service sectors. On the one hand, health services are not new to humanity. Hospitals are necessary in any country and have always existed. Similarly, sustainability issues are of global concern. On the other hand, although the framework was developed from the hospital perspective, this does not prevent it from being used as a reference for other service sectors, if properly adapted. Just as it is extremely important for hospitals to adopt supply chain management concepts and practices that have proved successful in other sectors such as food, research focused on hospitals can be a source of learning [13].

The proposed framework is composed of two main blocks: practices and performance, whose corresponding exploded views are depicted in Figures 5 and 6. For practices, contributions that conceptualize the logistics management process and supply chain integration [3], health care operations management [123], and hospital logistics [124] were considered. Accordingly, components traditionally related to internal supply chains, namely purchasing, warehousing and inventory, and transportation and distribution management, are placed at the center, and serve the care units through which patients flow, which include emergency, outpatient, diagnostic and therapy, operating theater, and inpatient [123]. Undirected arcs connect these units, which means the multiple directions in which patient flow occurs, since varied medical needs create customized sequences [125]. Clothes and laundry management as well as general services management are included, along with medical goods, food, energy, water, and waste management, since they are considered hospital logistic fields [124], account for resource consumption, and influence healthcare delivery, quality of patient care, and patient satisfaction [123]. In addition, strategic management and leadership, as well as information flow and technology management are key constituents of the framework, as they can influence and support supply chain relationships.

Upstream and downstream linkages are represented by Supplier Relationship Management (SRM) and Customer Relationship Management (CRM), respectively. Patient demand heads the list of inputs, as internal operations depend thereupon [123], and healthcare demand has unique characteristics. Rather than desire, healthcare services are grounded on necessity [126], which implies that typical marketing approaches to stimulate demand are minimal, if at all applicable, in healthcare. Other framed inputs include suppliers, medical and non-medical supplies and equipment, staff, other hospitals and providers, health insurers, government agencies, and regulators, which is in line with previous healthcare operations definitions and the numerous players that provide goods, services, and information to make operations possible [123,127]. Regarding outputs, these comprise the health status that reflects in clinical indicators, the client perception that indicates how well staff and patient expectations are met, and the use of resources that denote operation efficiency [123].

In accordance with the scope of the CRM concept, identified practices in the reviewed literature may fall short. It was struggling to identify it as a clear construct because these practices are scarce in healthcare [128], despite the fact that their influence on performance has been widely studied in manufacturing [5,7,129–131]. In such publications, CRM practices have been operationalized into management of customer complaints, evaluation of customer satisfaction, determination of customer expectations, frequent interaction with customers to set standards, and consideration of information from customers for business design and planning. To fill the existing void in healthcare, Reference [128] emphasizes the need to fortify the adoption of CRM practices, considering that they can lead to better understanding of patient profitability, and that there is some evidence of their contribution to patient health and loyalty.

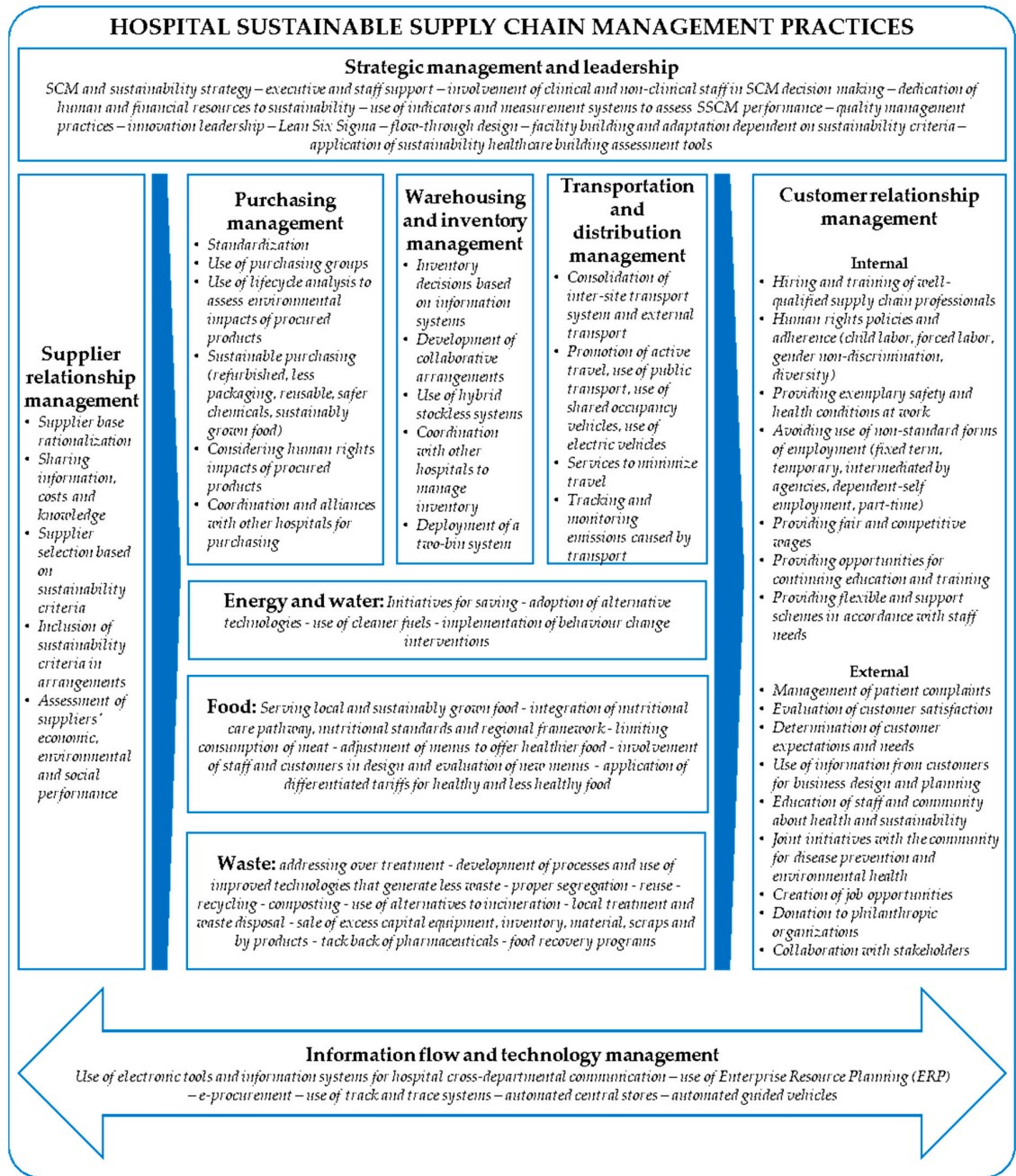

**Figure 5.** Conceptual framework for hospital sustainable supply chain management practices.

Recent studies on SSCM and specific social healthcare problems also lead to the inclusion of additional practices in the proposed framework. On the one hand, among the issues addressed by employee-centered social practices are wages, worker safety, and occupational health working conditions, employee participation, career planning for staff development, and the provision of opportunities for employees continuing their education [33,39,42,105,106,132–134]. On the other hand, community-centered social practices encompass labor laws, no child labor and human rights compliance, environmental awareness training, promotion of corporate social responsibility in the industry, sustainability reporting, donation to philanthropic organizations, provision of employment or business opportunities to the surrounding community, support of local health, educational, and cultural development, and volunteering at local charities [33,105,106,133,135].

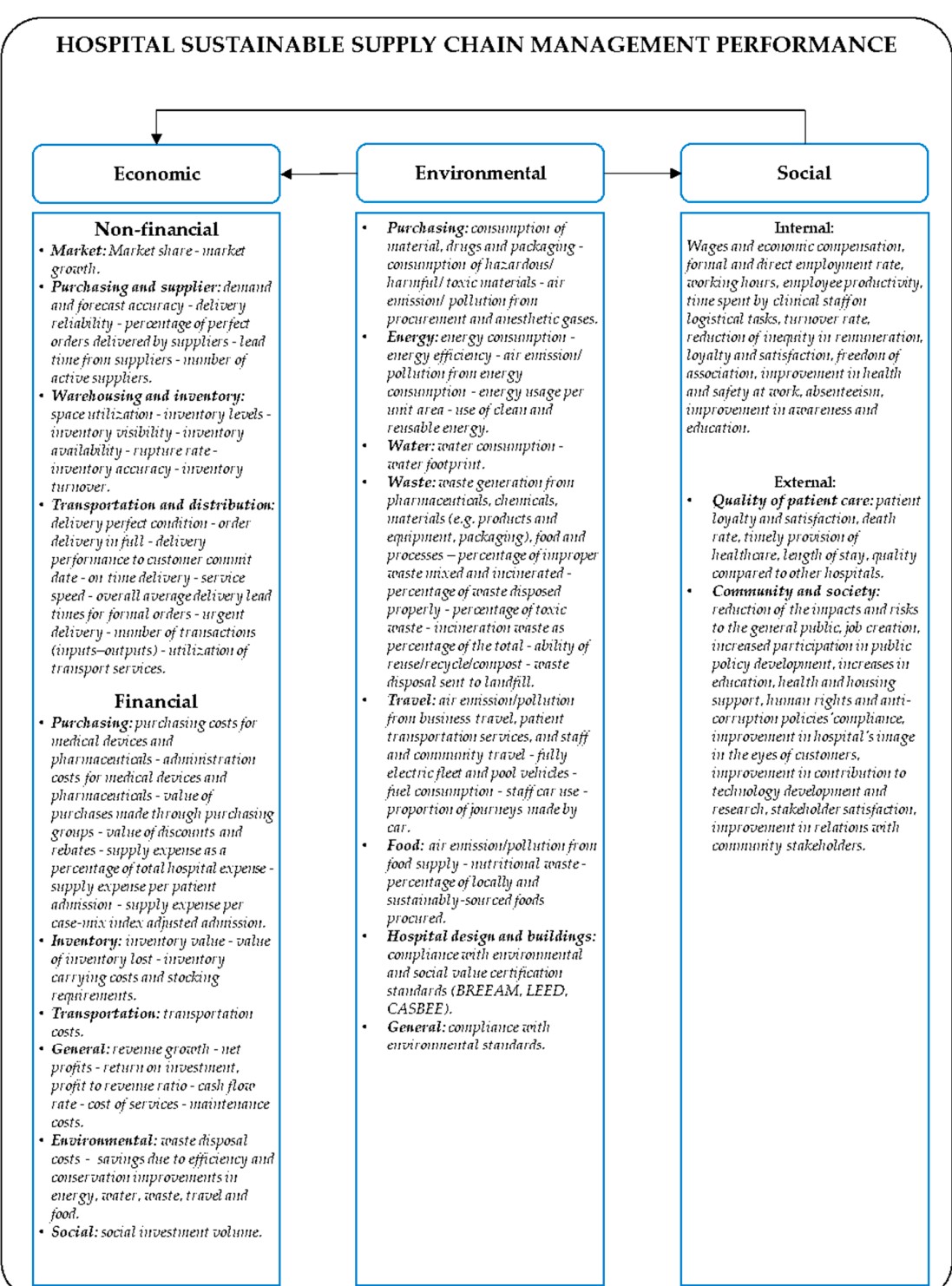

**Figure 6.** Conceptual framework for hospital sustainable supply chain management performance.

In alignment with the TBL approach, performance components are framed in terms of economic, environmental, and social dimensions. From both non-financial and financial perspectives, economic metrics point to measure operational issues and the costs of classical logistics processes such as purchasing, warehousing, inventory, transportation, and distribution management. However, market-specific metrics are also included in non-financial metrics, since they were reiterative in the reviewed literature. Analogously, general metrics intended to reflect outcomes of the entire hospital,

such as profitability, cost of services, and return on investment, are included in financial metrics. Moreover, environmental performance is sometimes converted into financial terms by quantifying actual and potential savings due to efficiency and conservation improvements in energy, water, travel, food, and waste [19,50,58,59,86,88,91,92,98], while social investment volume is used to economically measure social performance [72].

Environmental performance metrics are classified in accordance with the topics of purchasing, energy, water, travel, food, hospital design, and waste. To a certain extent, it could be said that carbon emissions reduction is the last target of environmental interventions, since it depends upon other metrics such as the reduction of resources, materials, drugs, and packaging consumption, increases in the use of clean and renewable energy, decreases in car use, percentage of locally and sustainably sourced procured foods, and avoidance of improper waste mixing. For example, it has been found that 5% of the carbon footprint of acute organizations comes from anesthetic gases [136]. Similarly, waste incinerators emit toxic air pollutants, such as dioxin and mercury [19,98,99]. Therefore, emissions can be reduced as consumption and waste generation decrease.

Except a few data regarding percental decrease in injuries caused by improper disposal [19] and improvements in awareness, education, and efficiency [86], hospital social performance metrics are scarce or vaguely addressed in the reviewed literature. For example, Reference [72] mentions employee satisfaction, work life quality, proportion of working hours to those planned, staff absenteeism, and employee privacy as concepts of hospital SCM, without distinguishing between practices and performance variables. Thus, contributions that focus on SSCM and manufacturing settings are taken as a basis for the framework [32,43,105,106,137–140]. In particular, metrics are divided into internal and external ones, and the latter, in turn, into community and society [137].

Metrics derived from the quality of patient care are included in the social external perspective, since hospital supply chains are social by nature, and failures in service provision may have fatal consequences on health and life [18]. Reference [60] found significant differences between leading, developing, and under-developed hospitals, in terms of performance on quality indicators, which means that a lower death rate and higher timely provision of healthcare are perceived by those hospitals with the greatest extent of applying healthcare SCM practices. In the same way, quality of patient care has been operationalized in terms of patient experience criteria [72,86,101] and perceptions in comparison with other hospitals [30,83].

In the proposed framework, relationships between sustainable performance dimensions are drawn. First, it is noted that the environmental dimension influences the economic and the social dimensions. Second, the social dimension influences the economic dimension. Those cases reported by References [19,78,87] are only a few of the studies that highlight specific economic outcomes of implementing initiatives through which the consumption of resources, such as energy and water, is reduced. According to Reference [58], improved environmental performance prevents health systems from incurring costs, and positively impacts the social dimension by reducing diseases caused by climate change.

Lastly, the economic dimension is thought to be influenced by the social one. Despite the lack of financial indicators that reflect the management of social issues in the supply chain, a significant number of studies (albeit not focused on hospitals) have concluded that social performance positively impacts economic performance [110]. In particular, Reference [110] argues that the implementation of SCM practices that seek to enhance social issues results in greater loyalty, legitimacy, socially responsible investment, and trust, as well as in lower stakeholder criticisms and risk, which, in turn, lead to cost reduction and increased economic benefits.

## 5. Further Research Agenda

In accordance with the reviewed literature, avenues worthy of future research comprise both methodological and conceptual issues. Most suggestions are concerned with limitations of generalizability, research methods, and scope.

- **Generalizability.** Some contributions recommend using wider samples [50,60,67,92] and replicating studies in other cities and countries [16,60,61,64,69,83,92,96,101], since more information from different populations and geographical areas might help validate existing research and explain heterogeneities. The broadening of moderating variables is also emphasized. Reference [141], for instance, found that different priorities are held by public and private hospitals in terms of sustainability dimensions, since pressures undergone appear to be dissimilar for both organization types. Apart from hospital type and size [67,83], suggested moderators include operations outsourcing [83], information applications by type [67], forms of technology [63], nature of purchases [69], and contingent factors that affect the inventory [79].

- **Research methods.** Directions for the research methods employed depend largely on the types of studies covered in the reviewed literature. For instance, papers with an analytical and mathematical foci advocate addressing parameters that allow the simplification and improvement of proposed models [75,84,87]. Similarly, other studies posit that qualitative data is desirable to complement quantitative results [60], whereas those based on qualitative data require empirical validation through quantitative tools, as mentioned by Reference [69]. Moreover, some researchers point out the limitations of cross-sectional studies, and, therefore, recommend the use of longitudinal designs, in order to learn about supply chain relationships over time [69], and to unveil the effects of these practices on performance in the long run [39,83]. Ultimately, the concept of being sustainable implies a long-term vision and a strategic approach [39].

- **Scope.** The need to dig deeper into what is meant by hospital SSCM practices and their influence on sustainable performance is brought to light in several ways. Technological, clinical, and organizational innovations that help hospitals be more sustainable are bound to being more explored [50]. In addition, the documentation of less successful practices, in contrast with the most successful ones, is stressed as an issue that needs additional attention [61], albeit more dissemination of exemplar cases is also required to encourage the adoption of practices [89]. Furthermore, much can be said about the impacts of hospital supply chains, but the measurement of the effects themselves represents a challenge for hospitals. As Reference [61] found, few hospitals use a wide range of indicators for purchase and inventory management. Reference [63] recommends including patient safety as a performance dimension. From an environmental standpoint, Reference [50] highlights the measurement of footprints across internal hospital supply chains as imperative.

In addition, further analysis of the influences of practices on performance is outlined. It is important to disclose the ways in which specific practices affect specific performance indicators [60], at the time that the incorporation of sectorial, social, and cultural issues into hospital SSCM research becomes prominent. While it can be a good practice to hire and train well-qualified supply chain professionals [17], it can be equally vital to know which concrete skills are required by supply chain managers in hospital settings [16]. While the relevance of promoting active travel is almost indisputable, the determinants of travel behavior remain unclear [50]. While adjusting menus to offer healthier dishes in hospital cafeterias is urgent, preference for less healthy food is rooted in the mindsets of the majority [91]. Consequently, since social and cultural factors can hinder or facilitate the implementation of practices [89], it is of paramount importance for sustainability improvement to gain understanding about the ways in which behaviors and culture need to change [50], and what kind of incentives and motivations lead staff and communities to demand, adopt, and promote better practices [91,93].

Lastly, the extant need of additional integrative research on hospital SSCM and sustainable performance merits mention. Most studies address the economic dimension, whereas few address the environmental one, and fewer yet address the social one. Unfortunately, although the under-representativeness of the social dimension is not unusual in the field of SCM [109], and it is difficult to ignore the economic rationality on which SCM research is based, it is clear that hospitals have social concerns that, if ignored, will make a growing healthcare deterioration more evident. This is more than serious, which takes into account the interdependence between health and sustainable

development, since one of the goals of sustainable development is oriented toward health improvement, but health is a condition for sustainable development [14].

## 6. Conclusions

Framing both SSCM practices and sustainable performance metrics at once is not an easy task. The concept of practice, per se, is difficult to define. Practices take various forms and can represent technologies, processes, ways of doing things, or ways of organizing work [65,68]. In addition, they can have different meanings or rationales from a sustainability approach and, for this reason, can overlap whichever categories have been established for their classification. In this way, a practice such as serving locally grown food can be conceived to improve food freshness and nutritional quality, favor the environment by avoiding transport activities, or strengthen local economies. Multiple purposes and interconnections among practices are more than visible and demonstrate the massive opportunities for action and impact that an integrated approach for sustainability provides, as well as its complexity.

Regarding performance, the main difficulty is that many effects of practices are not completely clear because there have not been enough empirical studies completed, and even less so regarding the interactions and trade-offs that may arise between dimensions. Moreover, indicator operationalization and validation are still incipient. For instance, not all the items encompassed by the review and the proposed framework have been measured in the literature. On one hand, it might be indicative of the exploratory status of current research, and the nascent interest in disclosing the elements that make up hospital SSCM. On the other, this could be interpreted as a symptom of the low level of adopting metrics and measurement systems, to such an extent that it would be more important to learn whether hospitals use indicators to measure performance than to calculate the values of such indicators.

The proposed framework can serve as a starting point for studying SSCM practices implementation in hospitals, in order to improve performance in this type of organizations, from a holistic sustainability approach. However, it needs to be validated and refined by using both quantitative and qualitative research methods. The practices and performance metrics covered are examples extracted from the literature to allow for a complete overview, rather than an instruction manual to be followed uncritically, since hospitals vary in accordance with their range of services, capacities, types, complexities, technologies, problems, impacts, needs, and more. Furthermore, it would be useful to prioritize elements of the framework, such as through multiple-criteria decision analysis techniques.

Apart from further validation required of the proposed framework, this review has several limitations. Additional databases and languages could be used. Since a search strategy that separately includes each of the topics of SCM or sustainability in hospitals was not formulated, the resulting analysis is comprehensive, but leaves room for improvements in exhaustiveness. The identification of categories of practices and performance could be an input with which to carry out a more thorough, detailed search for evidence, and enhance forthcoming debates on existing relationships. In addition, an interesting way to refine the definition of SSCM practices could be by covering literature that addresses drivers, barriers, and enablers. These were not fully or directly considered in the paper at hand, due to the early development of the proposed framework, but these could delineate a way for, or even help to explain which practices are or should be adopted, and why. Another limitation refers to subjectivity regarding the selection of keywords and paper classification, as well as in terms of established categories for practices and performance metrics, despite three researchers that have been involved throughout the review process. Recognized methodology guidelines have been referred to and followed.

**Author Contributions:** Conceptualization, V.D.-U., W.S., and E.V.G. Methodology, V.D.-U., W.S., and E.V.G. Writing—original draft preparation, V.D.-U. Writing—review and editing, V.D.-U., W.S., and E.V.G. Supervision, W.S. and E.V.G.

**Funding:** This research received no external funding.

**Acknowledgments:** This work is part of the doctoral study of the first author, which is financially supported by the Departamento Administrativo de Ciencia, Tecnología e Innovación – COLCIENCIAS, Colombia.

**Conflicts of Interest:** The authors declare no conflict of interest.

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
