# Peer review of "Sustainable Supply Chain Management Practices and Sustainable Performance in Hospitals: A Systematic Review and Integrative Framework"

_sustainability, doi:10.3390/su11215949_

Round 1

Reviewer 1 Report

I see your study as interesting literature/conceptual research, (1) first of all improve your reference list, which needs to be alphabetical, (2) try to specify more clearly the originality of your concept, referred to other service sector for example as known from literature, and applicability in any country, in the discussion part (3) refer more to this originality in abstract and less to your methodology details - note pls that abstract is very important to your study impact/popularity...

Author Response

Dear Reviewer:

We express our sincere appreciation for your valuable comments on the 616130 manuscript. We have carefully considered your comments and suggestions and attempted to properly address all comments. A point-by-point list of our responses to your valuable recommendations is included below. Modifications and changes contained in the revised manuscript are highlighted in yellow.

Comment 1: First of all improve your reference list, which needs to be alphabetical.

Response: According to the guidelines established by the journal, “References must be numbered in order of appearance in the text (including table captions and figure legends) and listed individually at the end of the manuscript”. Therefore, it is not possible for us to present the reference list in alphabetical order. We hope the reviewer comprehension.

Comment 2: Try to specify more clearly the originality of your concept, referred to other service sector for example as known from literature, and applicability in any country, in the discussion part.

Response: In section four, three paragraphs were included to specify more clearly the contribution of the proposed framework. The integration of practices and performance considering the three sustainability dimensions were included. The level of operationalization of practices and performance metrics was also mentioned, as well as the potential applicability in different regions and other service sectors.

Comment 3: Refer more to this originality in abstract and less to your methodology details - note pls that abstract is very important to your study impact/popularity...

Response: The abstract was modified in order to improve clarity, highlight the contribution of the paper, and reduce emphasis on methodology details.

Best regards.

Reviewer 2 Report

Authors present a interesting study about sustainable supply chain management practices to sustainable performance in hospital setting. This is an interesting paper but it is necessary some changes in the document:

.- Abstract: The abstract is too confused. Abstract must contain background for provides a rationale for the study and states the main aim. In addition, it must contain results for describes the main findings (with important numerical values, if this is possible.) and conclusions. Conclusions including the main results and advances the field of study.

.- Figure 1 does not provide relevant information. The information provided by the Figure is on lines 110 to 113. The authors must delete Figure 1.

.- Figure 2: The information included in Figure 2 is redundant, it is already included in Table 1. Please remove Figure 2.

Section 4 "A proposed framework for hospital SSCM" has a high quality and very interesting reflections. Congratulations .

Section 5 is very extensive. The authors must justify the existence of section 5. This reviewer believes that its inclusion is not necessary.

Author Response

Dear Reviewer:

We express our sincere appreciation for your valuable comments on the 616130 manuscript. We have carefully considered your comments and suggestions and attempted to properly address all comments. A point-by-point list of our responses to your valuable recommendations is included below. Modifications and changes contained in the revised manuscript are highlighted in yellow.

Comment 1: Abstract: The abstract is too confused. Abstract must contain background for provides a rationale for the study and states the main aim. In addition, it must contain results for describes the main findings (with important numerical values, if this is possible.) and conclusions. Conclusions including the main results and advances the field of study.

Response: The abstract was modified in order to improve clarity. A more structured order was followed considering the sequence of topic, gap, main aim, results, contribution and conclusions. Both results and conclusions were mentioned in more detail.

Comment 2: Figure 1 does not provide relevant information. The information provided by the Figure is on lines 110 to 113. The authors must delete Figure 1.

Response: Figure 1 was removed.

Comment 3: Figure 2: The information included in Figure 2 is redundant, it is already included in Table 1. Please remove Figure 2.

Response: The contents of Figure 2 and Table 1 were examined. Figure 2 is included in the materials and methods section and comprises the stages and steps followed for the literature review, while Table 1 is included in the results section and summarizes the categories and examples of identified sustainable supply chain management practices in hospitals. Due to the information is not redundant, we considered to maintain the Figure 2.

Comment 4: Section 4 "A proposed framework for hospital SSCM" has a high quality and very interesting reflections. Congratulations.

Response: We are very grateful for your kind comment.

Comment 5: Section 5 is very extensive. The authors must justify the existence of section 5. This reviewer believes that its inclusion is not necessary.

Response: We really appreciate this suggestion. After analyzing it carefully, it was considered important to keep section 5, as one of the reasons for performing systematic reviews is “to identify any gaps in current research in order to suggest areas for further investigation” (Kitchenham, 2004, pp. 2). In this sense, the literature review, that is part of the submitted manuscript, allows to identify avenues worthy of future research that may be of interest to academics who focus on sustainability and supply chain management, among other fields of study.

Best regards.

Reviewer 3 Report

The document is complete and makes a detailed literature review. Please avoid using personal pronouns, such as we, our, among others. Figure 4 must have the caption below it. Currently the caption is in the top. In Table 1, in my opinion, is better if for every Examples of practices, some references are associated to them. Currently, the references are associated to categories and not to Examples of practices. Table 1 appears twice. The Table on page 9 should be number 2. Renumber subsequent tables. In many paragraphs, up to three references appear without a discussion of them. Please discuss the references. 278 articles were identified, of which 150 were removed, please indicate the principles for inclusion and exclusion. In Table 1, in the category of Other practices, appears waste elimination. Why is it not in the Waste management category? Review if there are no repeated practices in the different categories of Table 1.

Author Response

Dear Reviewer:

We express our sincere appreciation for your valuable comments on the 616130 manuscript. We have carefully considered your comments and suggestions and attempted to properly address all comments. A point-by-point list of our responses to your valuable recommendations is included below. Modifications and changes contained in the revised manuscript are highlighted in yellow.

Comment 1: Please avoid using personal pronouns, such as we, our, among others.

Response: Modifications were addressed to avoid using personal pronouns.

Comment 2: Figure 4 must have the caption below it. Currently the caption is in the top.

Response: We apologize for this mistake. The caption of Figure 4 (Figure 3 now) was located below.

Comment 3: In Table 1, in my opinion, is better if for every Examples of practices, some references are associated to them. Currently, the references are associated to categories and not to Examples of practices.

Response: We really appreciate this suggestion. References were associated to examples of practices, instead of categories.

Comment 4: Table 1 appears twice. The Table on page 9 should be number 2. Renumber subsequent tables.

Response: We apologize for this mistake. The numbering of the tables and figures was verified and corrected.

Comment 5: In many paragraphs, up to three references appear without a discussion of them. Please discuss the references.

Response: We really appreciate this suggestion. The discussion of references was enhanced throughout the paper.

Comment 6: 278 articles were identified, of which 150 were removed, please indicate the principles for inclusion and exclusion.

Response: Inclusion and exclusion criteria were specified in the materials and methods section. To facilitate the reading of the section, subtitles related to the planning stage were added.

Comment 7: In Table 1, in the category of Other practices, appears waste elimination. Why is it not in the Waste management category? Review if there are no repeated practices in the different categories of Table 1.

Response: We really appreciate this suggestion. After analyzing it carefully, practices related to lean supply chain management were relocated in the waste management category, while quality management practices and patient flow logistics, remained in the category of other practices.

Best regards.

Round 2

Reviewer 2 Report

Thanks for attending the recommendations and improve the quality of the document. For me, the paper is ready.